# MODELING QUESTION ASKING USING NEURAL PROGRAM GENERATION

## ABSTRACT

People ask questions that are far richer, more informative, and more creative than current AI systems. We propose a neural program generation framework for modeling human question asking, which represents questions as formal programs and generates programs with an encoder-decoder based deep neural network. From extensive experiments using an information-search game, we show that our method can ask optimal questions in synthetic settings, and predict which questions humans are likely to ask in unconstrained settings. We also propose a novel grammar-based question generation framework trained with reinforcement learning, which is able to generate creative questions without supervised data.

## 1 INTRODUCTION

People can ask rich, creative questions to learn efficiently about their environment. Question asking is central to human learning yet it is a tremendous challenge for computational models. There is always an infinite set of possible questions that one can ask, leading to challenges both in representing the space of questions and in searching for the right question to ask.

Machine learning has been used to address aspects of this challenge. Traditional methods have used heuristic rules designed by humans (Heilman & Smith, 2010; Chali & Hasan, 2015), which are usually restricted to a specific domain. Recently, neural network approaches have also been proposed, including retrieval methods which select the best question from past experience (Mostafazadeh et al., 2016) and encoder-decoder frameworks which map visual or linguistic inputs to questions (Serban et al., 2016; Mostafazadeh et al., 2016; Yuan et al., 2017; Yao et al., 2018). While effective in some settings, these approaches do not consider settings where the questions are asked about partially unobservable states. Besides, these methods are heavily data-driven, limiting the diversity of generated questions and requiring large training sets for different goals and contexts. There is still a large gap between how people and machines ask questions.

Recent work has aimed to narrow this gap by taking inspiration from cognitive science. For instance, Lee et al. (2018) incorporates aspects of "theory of mind" (Premack & Woodruff, 1978) in question asking by simulating potential answers to the questions, but the approach relies on imperfect agents for natural language understanding which may lead to error propagation. Related to our approach, Rothe et al. (2017) proposed a powerful question-asking framework by modeling questions as symbolic programs, but their algorithm relies on hand-designed program features and requires expensive calculations to ask questions.

We use "neural program generation" to bridge symbolic program generation and deep neural networks, bringing together some of the best qualities of both approaches. Symbolic programs provide a compositional "language of thought" (Fodor, 1975) for creatively synthesizing which questions to ask, allowing the model to construct new ideas based on familiar building blocks. Compared to natural language, programs are precise in their semantics, have clearer internal structure, and require a much smaller vocabulary, making them an attractive representation for question answering systems as well (Johnson et al., 2017; Yi et al., 2018; Mao et al., 2019). However, there has been much less work using program synthesis for question asking, which requires searching through infinitely many questions (where many questions may be informative) rather than producing a single correct answer to a question. Deep neural networks allow for rapid question-synthesis using encoder-decoder modeling, eliminating the need for the expensive symbolic search and feature evaluations in Rothe et al. (2017). Together, the questions can be synthesized quickly and evaluated formally for quality

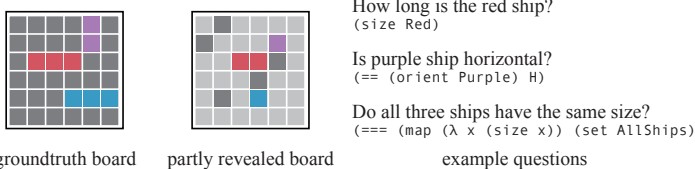

groundtruth board    partly revealed board        example questions

Figure 1: The Battleship task. Blue, red, and purple tiles are ships, dark gray tiles are water, and light gray tiles are hidden. The agent can see a partly revealed board, and should ask a question to seek information about the hidden board. Example questions and translated programs are shown on the right. We recommend viewing the figures in color.

(e.g. the expected information gain), which as we show can be used to train question asking systems using reinforcement learning.

In this paper, we develop a neural program generation model for asking questions in an information-search game similar to "Battleship" used in previous work (Gureckis & Markant, 2009; Rothe et al., 2017; 2018). The model uses a convolutional encoder to represent the game state, and a Transformer decoder (Vaswani et al., 2017) for generating questions. Building on the work of Rothe et al. (2017), the model uses a grammar-enhanced question asking framework, such that questions as programs are formed through derivation using a context free grammar. Importantly, we show that the model can be trained from human demonstrations of good questions using supervised learning, along with a data augmentation procedure that leverages previous work to produce additional human-like questions for training. Our model can also be trained without such demonstrations using reinforcement learning. We evaluate the model on several aspects of human question asking, including reasoning about optimal questions in synthetic scenarios, density estimation based on free-form question asking, and creative generation of genuinely new questions.

To summarize, our paper makes three main contributions: 1) We propose a neural network for modeling human question-asking behavior, 2) We propose a novel reinforcement learning framework for generating creative human-like questions by exploiting the power of programs, and 3) We evaluate different properties of our methods extensively through three different experiments.

## 2 RELATED WORK

Question generation has attracted attention from the machine learning community. Early research mostly explored rule-based methods which strongly depend on human-designed rules (Heilman & Smith, 2010; Chali & Hasan, 2015). Recent methods for question generation adopt deep neural networks, especially using the encoder-decoder framework, and can generate questions without hand-crafted rules. These methods are mostly data-driven, which use pattern recognition to map inputs to questions. Researchers have worked on generating questions from different types of inputs such as knowledge base facts (Serban et al., 2016), pictures (Mostafazadeh et al., 2016), and text for reading comprehension (Yuan et al., 2017; Yao et al., 2018). However aspects of human question-asking remain beyond reach, including the goal-directed and flexible qualities that people demonstrate when asking new questions. This issue is partly addressed by some recent papers which draw inspiration from cognitive science. Research from Rothe et al. (2017) and Lee et al. (2018) generate questions by sampling from a candidate set based on goal-oriented metrics. This paper extends the work of Rothe et al. (2017) to overcome the limitation of the candidate set, and generate creative, goal-oriented programs with neural networks.

Our work also builds on neural network approaches to program synthesis, which have been applied to many different domains (Devlin et al., 2017; Sun et al., 2018; Tian et al., 2019). Those approaches often draw inspiration from computer architecture, using neural networks to simulate stacks, memory, and controllers in differentiable form (Reed & De Freitas, 2016; Graves et al., 2014). Other models incorporate Deep Reinforcement Learning (DRL) to optimize the generated programs in a goal oriented environment, such as generating SQL queries which can correctly perform a specific database processing task (Zhong et al., 2018), translating strings in Microsoft Excel sheets (Devlin et al., 2017), understanding and constructing 3D scenes (Liu et al., 2019) and objects (Tian et al., 2019). Recent work has also proposed ways to incorporate explicit grammar information into the program synthesis process. Yin & Neubig (2017) design a special module to capture the grammar information as a prior, which can be used during generation. Some recent papers (Bunel et al., 2018;

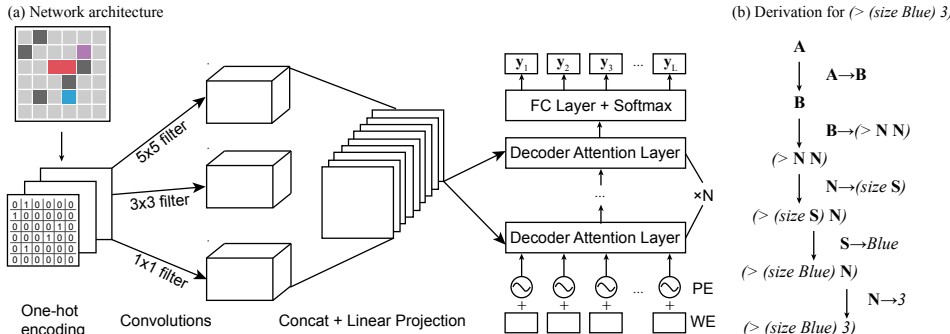

Figure 2: Neural program generation. Figure (a) shows the network architecture. The board is represented as a grid of one-shot vectors and is embedded with a convolutional neural network. The board embedding and a sequence of symbols are inputted to a Transformer decoder (Vaswani et al., 2017) to generate output vectors (details in section 4). PE means positional embeddings, and WE means word embeddings. (b) shows the derivation steps for program " `(> (size Blue) 3)` " using CFG. Non-terminals are shown as bold-faced, and terminals are shown in italic. The production rules used are shown next to each arrow.

Si et al., 2019) encode grammar with neural networks and use DRL to explicitly encourage the generation of semantically correct programs. Our work differs from these in two aspects. First, our goal is to generate informative human-like questions in the new domain instead of simply correct programs. Second, we more deeply integrate grammar information in our framework, which directly generates programs based on the grammar.

## 3 BATTLESHIP TASK

In this paper, we work with a task used in previous work for studying human information search (Gureckis & Markant, 2009) as well as question asking (Rothe et al., 2018). The task is based on an information search game called "Battleship", in which a player aims to resolve the hidden layout of the game board based on the revealed information (Figure 1). There are three ships with different colors (blue, red, and purple) placed on a game board consisting of $6 \times 6$ grid of tiles. Each ship can be either horizontal or vertical, and takes 2, 3 or 4 tiles long. All tiles are initially turned over (light grey in Figure 1), and the player can flip one tile at a time to reveal an underlying color (either a ship color, or dark grey for water). The goal of the player is to determine the configuration of the ships (positions, sizes, orientations) in the least number of flips.

In the modified version of this task studied in previous work (Rothe et al., 2017; 2018), the player is presented with a partly revealed game board, and is required to ask a natural language question to gain information about the underlying configuration. As shown in Figure 1, the player can only see the partly revealed board, and might ask questions such as "How long is the red ship?" In this paper, we present this task to our computational models, and ask the models to generate questions about the game board.

Rothe et al. (2017) designed a powerful context free grammar (CFG) to describe the questions in the Battleship domain. The grammar represents questions in a LISP program-like format, which consists of a set of primitives (like numbers, colors, etc.) and a set of functions over primitives (like arithmetic operators, comparison operators, and other functions related to the game board). Another research by (Rothe et al., 2018) shows that it captures the full range of questions that people asked in an extensive dataset, mainly because the majority of this grammar is general functions which make it flexible enough. The grammar is able to generate an infinite set of other possible questions beyond collected human questions, capturing key notions of compositionality and computability. Figure 1 provides some examples of produced programs. The full grammar is provided in Appendix C.

## 4 NEURAL PROGRAM GENERATION FRAMEWORK

The neural network we use includes a Convolutional Neural Network (CNN) for encoding the input board, and a Transformer (Vaswani et al., 2017) decoder for estimating the symbol distribution or selecting actions in different settings described below. The input $x \in \{0, 1\}^{6 \times 6 \times 5}$ is a binary representation of the 6x6 game board with five channels, one for each color to be encoded as a one-hot vector in each grid location. A simple CNN maps the input $x$ to the encoder output $\mathbf{e} \in \mathbb{R}^{6 \times 6 \times M}$,

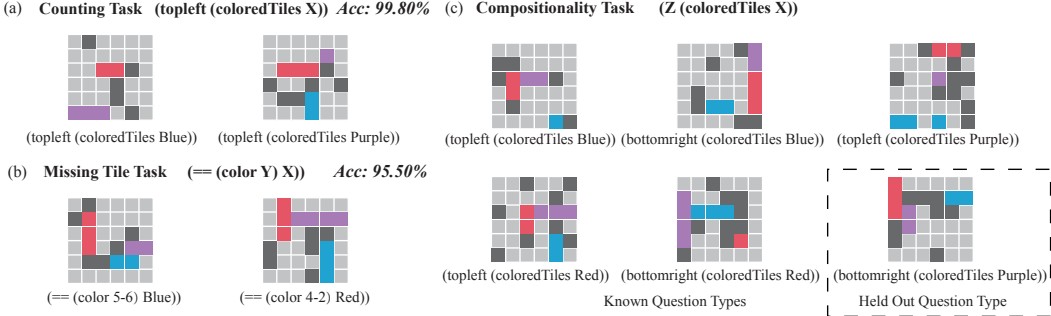

Figure 3: Design of the tasks in experiment 1. The goal of task (a) is to find the color which has the least number of visible tiles; the goal of task (b) to find the location and color of the missing tile; (c) is the compositionality task with 5 questions as known question types, and another one (in dotted box) as held out question type. The format of generated question is shown alongside the title of each task, where X, Y and Z are variables. The accuracy of supervised model for task (a) and (b) are given below each task.

where $M$ is the length of encoded vectors. Then a Transformer decoder takes $\mathbf{e}$ and a sequence of length $L$ as input, and outputs a sequence of vectors $\mathbf{y}_i \in \mathbb{R}^{N_o}, i = 1 \cdots L$, where $N_o$ is the output size. As shown later, the input sequence and output vectors can be interpreted differently in different settings. The model is shown in Figure 2(a), and details are described in Appendix A.

Our model is compatible with both supervised and reinforcement training.

**Supervised training.** In the supervised setting, the goal is to model the distribution of questions present in the training set. Each output $\mathbf{y}_i \in \mathbb{R}^{N_o}$ is a symbol at position $i$ in the program, where $N_o$ is the number of different symbols in the grammar. The model is trained with symbol-level cross entropy loss, and can be used to calculate the log-likelihood of a given sequence, or to generate a question symbol-by-symbol from left to right. Generation works as follows. Suppose at step $t$, a sequence of length $t$ along with the encoded board is presented to the decoder. The model predicts the vector $\mathbf{y}_t$ which represents the probability of each symbol to be chosen as next. Then we sample a symbol at location $t + 1$ and execute the decoder again with the new sequence, until an <eos> symbol is generated or maximum length is reached.

**Sequence-based RL.** The framework can be adapted to generate a sequence of symbols without stepwise supervision, such that reward is provided only after the entire question is generated.

**Grammar-enhanced RL.** Finally, the framework can be used with a novel grammar-enhanced RL training procedure. Figure 2(b) illustrates the process of generating a program from the context-free grammar specified in Rothe et al. (2017). Beginning from the start symbol "A", at each step a production rule is chosen and applied to one of the non-terminals in the current string. The choice of rule is modeled as a Markov Decision Process, and we solve it with DRL. Each state is a partially derived string passed to the decoder, and we use the first output $\mathbf{y}_1 \in \mathbb{R}^{N_o}$ to represent the probability of selecting each production rule from all possible $N_o$ rules. After the rule is applied, the new string is passed back into the decoder, repeating until only terminals are contained in the sequence. We adopt the leftmost derivation here to avoid the ambiguity of parsing order, so at each step the left-most non-terminal will be replaced.

## 5 EXPERIMENTS

### 5.1 REASONING IN SYNTHETIC SETTINGS

In the first experiment, we designed three tasks to evaluate whether the model can learn simple compositional rules and reasoning strategies. These tasks include counting the number of visible ship tiles, locating a missing ship tile, and generalizing both strategies to unseen scenario types. Figure 3 illustrates the three tasks we designed in this experiment by providing examples of each task.

### 5.1.1 TASK DESCRIPTIONS

**Counting task.** Models must select the ship color with the least number of visible tiles on the board. Each board has a unique answer, and models respond by generating a program " (topleft

Table 1: Accuracy (%) on the compositionality task using different numbers of training examples from the held out question type.

| # of training examples | 0 | 10 | 50 | 100 | 200 | 400 | 800 |
|---|---|---|---|---|---|---|---|
| Acc. on held out question type | 0.0 | 2.0 | 39.0 | 69.5 | 81.0 | 92.0 | 96.0 |
| Acc. on known question types | 96.6 | 97.3 | 97.1 | 96.0 | 96.3 | 97.8 | 96.1 |
| Acc. classify on held out question type | 33.0 | 37.0 | 49.0 | 75.5 | 88.0 | 94.0 | 99.5 |

(`coloredTiles X`)" where `X` is a ship color. $4000$ examples are used for training, and another $1000$ examples are used for testing.

**Missing tile task.** Models must select the ship that is missing a tile and identify which tile is missing. All ships are completely revealed except one, which is missing exactly one tile. Models respond by generating "(`== (color Y) X`)" where `X` is a color and `Y` is a location on the board. The number of training and test examples are the same as the counting task.

**Compositionality task.** Models must combine both of the above strategies to find the missing tile of the ship with the least visible tiles. Outputs are produced as "(`Z (coloredTiles X)`)" where `X` is a color and `Z` is either `topleft` or `bottomright`. Each board has a unique answer.

This task further evaluates compositionality by withholding question types from training. With three values for `X` and two for `Z`, there are six possible question types and one is picked as the "held out" type. The other five "known" question types have $800$ training examples. For the held out question type, the number of training examples is varied from $0$ to $800$, to test how much data is needed for generalization. Another $200$ new boards of each question type is used for evaluation. More information about the model hyperparameters and training procedures are provided in Appendix B.1.

### 5.1.2 RESULTS AND DISCUSSION

We train our model in a fully supervised fashion. Accuracy for the counting and missing tile tasks is shown in Figure 3. The full neural program generation model shows strong reasoning abilities, achieving high accuracy for both the counting and missing tile tasks, respectively. We also perform ablation analysis of the encoder filters of the model, and provide the results in Appendix D.

The results for the compositionality task are summarized in Table 1. When no training data regarding the held out question type is provided, the model cannot generalize to situations systematically different from training data, exactly as pointed out in previous work on the compositional skills of encoder-decoder models (Lake & Baroni, 2018). However, when the number of additional training data increases, the model quickly incorporates the new question type while maintaining high accuracy on the familiar question tasks. On the last row of Table 1, we compare our model with another version where the decoder is replaced by two linear transformation operations which directly classify the ship type and location (details in Appendix B.1). This model has $33.0\%$ transfer accuracy on compositional scenarios never seen during training. This suggests that the model has the potential to generalize to unseen scenarios if the task can be decomposed to subtasks and combined together.

### 5.2 ESTIMATING THE DISTRIBUTION OF HUMAN QUESTIONS

In this experiment, we examine if the neural network has the capability of capturing the distribution of human questions as a conditioned language model.

### 5.2.1 DATA AUGMENTATION

To train the model, we need to construct a training set of many paired game boards and questions. Instead of laboriously collecting a large number of real human questions, and translating them into programs by hand, we construct the dataset by sampling from a previous computational model of human question asking (Rothe et al., 2017). More precisely, we randomly generate a large number of game boards and sample $K$ questions given each board. For generating the boards, we first uniformly sample the configuration of three ships, and randomly cover arbitrary number of tiles, with the restriction that at least one ship tile is observable. Next we randomly sample $K$ programs for each board with importance sampling based on the cognitive model proposed by Rothe et al. (2017), which models the probability of a question under a given context as

$$p(q; \theta) = -\exp(\varepsilon(q; \theta))/Z \tag{1}$$

Table 2: Results of the second experiment.

(a) Log-likelihood (LL) on two evaluation sets (sampled data and human data) of different version models.

| Model | LL_sampled | LL_human |
|---|---|---|
| Full model | **-3.197** | **-7.124** |
| no pretrain | -3.217 | -7.280 |
| LSTM decoder | -3.222 | -9.013 |
| MLP encoder | -3.385 | -7.475 |
| decoder only | -3.401 | -8.434 |

(b) Log-likelihood (LL) on different split of sampled evaluation set based on the uncertainty of the board (called low entropy, medium entropy, high entropy respectively). More comparisons are provided in Appendix B Table 7.

| Model | LL_low | LL_mid | LL_high |
|---|---|---|---|
| Full model | **-2.990** | **-3.190** | **-3.414** |
| decoder only | -3.312 | -3.397 | -3.494 |

where $\varepsilon(\cdot)$ is a parameterized energy function for estimating the likelihood of a question being asked by human, which considers multiple features such as question informativeness, complexity, answer type, etc. $Z$ is a normalization constant.

We also randomly generate a larger set of questions to pretrain the decoder component of the model as a "language model" over questions, enabling it to better capture the grammatical structure of possible questions. Details regarding the model hyperparameters, training procedure, and pre-training procedure are provided in Appendix B.2.

### 5.2.2 RESULTS AND DISCUSSION

We evaluate the log-likelihood of reference questions generated by our full model as well as some lesioned variants of the full model, including a model without pretraining, a model with the Transformer decoder replaced by an LSTM decoder, a model with the convolutional encoder replaced by a simple MLP encoder, and a model that only has a decoder (unconditional language model). Though the method from Rothe et al. (2017) also works on this task, here we cannot compare with their method for two reasons. One is that our dataset is constructed using their method, so the likelihood of their method should be an upper bound in our evaluation setting. Additionally, they can only approximate the log-likelihood due to an intractable normalizing constant, and thus it difficult to directly compare with our methods.

Two different evaluation sets are used, one is sampled from the same process on new boards, the other is a small set of questions collected from human annotators. In order to calculate the log-likelihood of human questions, we use translated versions of these questions that were used in previous work (Rothe et al., 2017), and filtered some human questions that score poorly according to the generative model used for training the neural network (Appendix B.2).

A summary of the results is shown in Table 2a. The full model performs best on both datasets, suggesting that pretraining, the Transformer decoder, and the convolutional encoder are all important components of the approach. However, we find that the model without an encoder performs reasonably well too, even out-performing the full model with a LSTM-decoder on the human-produced questions. This suggests that while contextual information from the board leads to improvements, it is not the most important factor for predicting human questions. To further investigate the role of contextual information and whether or not the model can utilize it effectively, we conduct another analysis.

Intuitively, if there is little uncertainty about the locations of the ships, observing the board is critical since there are fewer good questions to ask. To examine this factor, we divide the scenarios based on the entropy of the hypothesis space of possible ship locations into a low entropy set (bottom 30%), medium entropy set (40% in the middle), and high entropy set (top 30%). We evaluate different models on the split sets of sampled data and report the results in Table 2b. When the entropy is high, it is easier to ask a generally good question like "how long is the red ship" without information of the board, so the importance of the encoder is reduced. If entropy is low, the models with access to the board has substantially higher log-likelihood than the model without encoder. Also, the first experiment (section 5.1) would be impossible without an encoder. Together, this implies that our model can capture important context-sensitive characteristics of how people ask questions.

### 5.3 QUESTION GENERATION

In this experiment, we evaluate our reinforcement learning framework proposed in Section 4 on its ability of generating novel questions from scratch, without providing a large training set.

Table 3: Evaluation results of experiment 3. Our grammar enhanced model is compared with a supervised trained baseline from experiment 2, a sequence generative RL baseline, and a text-based model. The models are compared in terms of average energy value, average expected information gain (EIG) value, the ratio of EIG value greater than 0.9/0, number of unique questions generated, and number of unique novel questions generated (by "novel" we mean questions not presented in the human dataset). The EIG of text-based model is calculated based on the program form of the generated questions.

| Model | avg. EIG | EIG>0.9 | EIG>0 | #unique | #unique novel |
|---|---|---|---|---|---|
| text-based | 0.928 | 62.80% | 76.95% | - | - |
| supervised | 1.033 | 51.65% | 84.55% | 137 | 9 |
| sequence RL | 1.235 | 58.60% | 75.20% | **167** | 52 |
| grammar enhanced RL | **1.266** | **84.75%** | **91.35%** | 141 | **129** |

The reward for training the reinforcement agent is calculated based on the energy value of the generated question $q$. We transform the energy value to a proper range for reward by $-\varepsilon(q)/10$ and clamp it between $-1$ and $1$. The model is optimized with the REINFORCE algorithm (Williams, 1992). A baseline for REINFORCE is established simply as the average of the rewards in a batch. In order to produce higher-quality questions, we manually tune the information-related parameter of the energy function from Rothe et al. (2017) to make it more information-seeking in this experiment. This process is described in Appendix B.2.

We compare the models on their ability to generate diverse questions with high expected information gain (EIG), which is defined as the expected reduction in entropy, averaged across all possible answers to a question $x$.

$$\text{EIG}(x) = \mathbb{E}_{d \in A_x}[I(p(h)) - I(p(h|d;x))] \tag{2}$$

where $I(\cdot)$ is the Shannon entropy. The terms $p(h)$ and $p(h|d;x)$ are the prior and posterior distribution of a possible ship configuration $h$ given question $x$ and answer $d \in A_x$. We compare our program-based framework with a simple text-based model, which has the same architecture but is trained with supervision on the text-based question dataset collected by (Rothe et al., 2017). We also compare with the supervised program-based model from the last experiment. Finally, we implement a sequence-based reinforcement learning agent that specifies the program without direct access to the grammatical rules. For this alternative RL agent, we find it necessary to pretrain for 500 epochs with stepwise supervision.

### 5.3.1 RESULTS AND DISCUSSION

The models are evaluated on 2000 randomly sampled boards, and the results are shown in Table 3. Note that any ungrammatical questions are excluded when we calculate the number of unique questions. First, when the text-based model is evaluated on new contexts, 96.3% of the questions it generates were included in the training data. We also find that the average EIG and the ratio of EIG>0 is worse than the supervised model trained on programs. Some of these deficiencies are due to the very limited text-based training data, but using programs instead can help overcome these limitations. With the program-based framework, we can sample new boards and questions to create a much larger dataset with executable program representations. This self-supervised training helps to boost performance, especially when combined with grammar-enhanced RL.

From the table, the grammar-enhanced RL model is able to generate informative and creative questions. It can be trained from scratch without examples of human questions, and produces many novel questions with high EIG. In contrast, the supervised model rarely produces new questions beyond the training set. The sequence-level RL model is also comparatively weak at generating novel questions, perhaps because it is also pre-trained on human questions. It also more frequently generates ungrammatical questions.

We also provide examples in Figure 4 to show the diversity of questions generated by the grammar enhanced model, and more in the supplementary materials. Figure 4a shows novel questions the model produces, which includes clever questions such as "Where is the bottom right of all the purple and blue tiles?" or "What is the size of the blue ship minus the purple ship?", while it can also sometimes generates meaningless questions such as "Is the blue ship shorter than itself?" Additional examples of generated questions are provided in Appendix B.

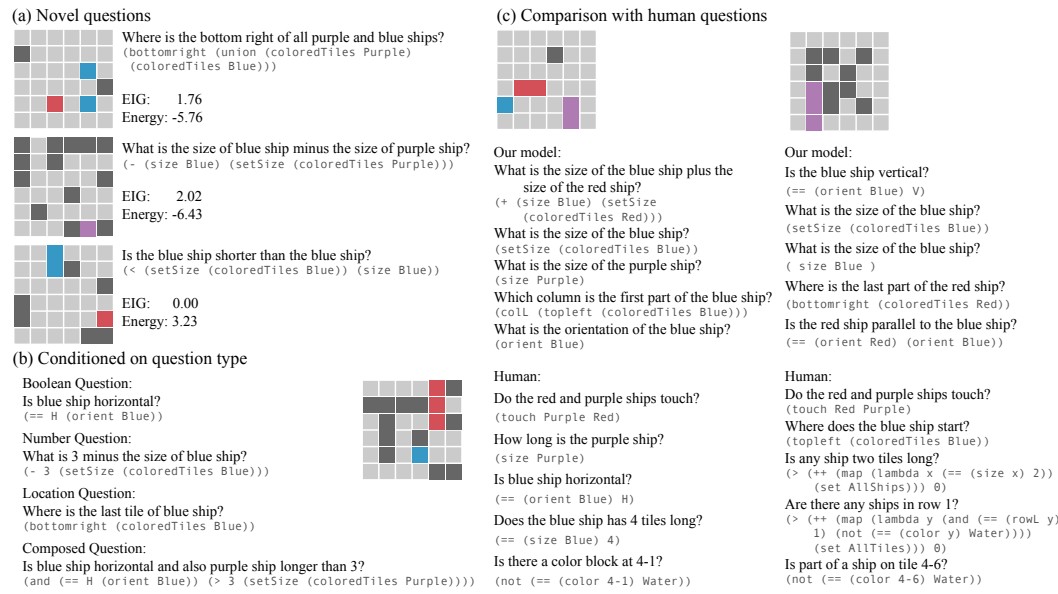

Figure 4: Examples of model-generated questions. The natural language translations of the question programs are provided for interpretation. (a) shows three novel questions generated by the grammar enhanced model, (b) shows an example of how the model generates different type of questions by conditioning the input to the decoder, (c) shows questions generated by our model as well as human annotators.

With the grammar enhanced framework, we can also guide the model to ask different types of questions, consistent with the goal-directed nature and flexibility of human question asking. The model can be queried for certain types of questions by providing different start conditions to the model. Instead of starting derivation from the start symbol "A", we can start derivation from a intermediate state such as "B" for Boolean questions or a more complicated "(and B B)" for composition of two Boolean questions. In Figure 4b, we show examples where the model is asked to generate four specific types of questions: true/false questions, number questions, location-related questions, and compositional true/false questions. We see that the model can flexibly adapt to new constraints and generate meaningful questions.

In Figure 4c, we compare the model generated questions with human questions, each randomly-sampled from the model outputs and the human dataset. These examples again demonstrate that our model is able to generate clever and human-like questions. However, we also find that people sometimes generate questions with quantifiers such as "any" and "all", which are operationalized in program form with lambda functions. These questions are complicated in representation and not favored by our model, showing a current limitation in our model's capacity.

## 6 CONCLUSION

We introduce a neural program generation framework for question asking task under partially un-observable settings, which is able to generate creative human-like questions with human question demonstrations by supervised learning or without demonstrations by grammar-enhanced reinforce-ment learning. Programs provide models with a "machine language of thought" for compositional thinking, and neural networks provide an efficient means of question generation. We demonstrate the effectiveness of our method in extensive experiments covering a range of human question asking abilities.

The current model has important limitations. It cannot generalize to systematically different scenarios, and it sometimes generates meaningless questions. We plan to further explore the model's compositional abilities in future work. Another promising direction is to model question asking and question answering jointly within one framework, which could guide the model to a richer sense of the question semantics. Besides, allowing the agent to iteratively ask questions and try to win the game is another interesting future direction. We would also like to use our framework in dialog systems and open-ended question asking scenarios, allowing such systems to synthesize informative and creative questions.

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

# A  NETWORK ARCHITECTURE

**Encoder.**  A simple CNN with one layer of filters is used to encode the board. Intuitively, many questions are related to specific positions, thus the position information should be recoverable from the encoding. On the other hand, some features of the board are translation-invariant, such as whether a ship is blocked by another ship. In order to capture the position-sensitive information as well as the translation-invariant patterns, three convolution operations with different filter sizes ($1 \times 1$, $3 \times 3$, and $5 \times 5$) are performed in parallel on the same input. The inputs are padded accordingly to make sure the feature maps have the same width and height. Then three feature maps are concatenated together along the dimension of output channels, and passed through a linear projection.

Formally, the outputs of the convolutions $\mathbf{c}$ can be obtained by

$$\mathbf{c} = \text{ReLU}([\text{Conv}_1(x); \text{Conv}_3(x); \text{Conv}_5(x)]) \tag{3}$$

where $\text{Conv}_k$ denotes a convolution operation on a $k \times k$ filter, $\text{ReLU}(\cdot)$ means applying a ReLU activation, and $[A; B]$ means the concatenation of matrices $A$ and $B$. Then $\mathbf{c} \in \mathbb{R}^{6 \times 6 \times 3C_{out}}$ is projected to the encoder output $\mathbf{e} \in \mathbb{R}^{6 \times 6 \times M}$ by matrix $W_o^e \in \mathbb{R}^{3C_{out}, M}$, where $C_{out}$ is the number of out channels of each convolution, and $M$ is the length of encoded vectors.

**Decoder.**  We use the decoder from the Transformer model (Vaswani et al., 2017). With an input sequence of length $L$, the decoder computes the hidden states through several stacked Decoder Attention Layers. Each layer is composed by three sub-layers, a self-attention module, an attention over the encoded board, and a fully connected feed-forward network. Residual connections are employed around each sub-layer, followed by a layer normalization (Ba et al., 2016). After $N$ layers of attention modules, a final output layer transforms the hidden states to the output vectors $\mathbf{y}_i \in \mathbb{R}^{N_o}$ at every position $i$ from 1 to $L$, where $N_o$ is the output size.

Given the output from encoder $\mathbf{e}$, and the hidden representation $\mathbf{h}^{n-1}$ from Decoder Attention Layer $n-1$, each layer computes the hidden representation as

$$\begin{aligned}
\mathbf{g}^n &= \text{LN}(\text{Self-ATT}(\mathbf{h}^{n-1}) + \mathbf{h}^{n-1}) \\
\mathbf{v}^n &= \text{LN}(\text{ATT}(\mathbf{g}^n, \mathbf{e}) + \mathbf{g}^n) \\
\mathbf{h}^n &= \text{LN}(\text{FC}(\mathbf{v}^n) + \mathbf{v}^n)
\end{aligned} \tag{4}$$

where $\text{LN}(\cdot)$ means layer normalization (Ba et al., 2016), $\text{FC}(\cdot)$ is a fully connected layer, $\text{ATT}(\cdot)$ and $\text{Self-ATT}(\cdot)$ are multi-head attention mechanisms, which computes the attention over the output of encoder $\mathbf{e}$, and the attention over the input $\mathbf{h}^{n-1}$ itself, respectively. They are defined as follows

$$\begin{aligned}
\text{ATT}(\mathbf{g}^n, \mathbf{e}) &= \text{Multi-ATT}(\mathbf{g}^n, \mathbf{e}, \mathbf{e}) \\
\text{Self-ATT}(\mathbf{h}^{n-1}) &= \text{Multi-ATT}(\mathbf{h}^{n-1}, \mathbf{h}^{n-1}, \mathbf{h}^{n-1})
\end{aligned} \tag{5}$$

$\text{Multi-ATT}(\cdot)$ is the multi-head attention mechanism described in the paper by Vaswani et al. (2017), which is a concatenation of multiple standard attention mechanisms with inputs projected using different matrices. A multi-head attention with $n$ heads is defined as

$$\begin{aligned}
\text{Multi-ATT}(Q, K, V) = W^o[\text{Attention}(W_1^Q Q, W_1^K K, W_1^V V); \cdots ; \\
\text{Attention}(W_n^Q Q, W_n^K K, W_n^V V)]
\end{aligned} \tag{6}$$

where

$$\text{Attention}(Q, K, V) = \text{softmax}(\frac{QK^T}{\sqrt{d_k}})V \tag{7}$$

is the scaled dot-product attention operation. $Q, K, V$ are a set of vectors called queries, keys, and values, respectively, and $d_k$ is the dimension of queries and keys.

After $N$ layers, we apply a linear projection and a softmax activation to $\mathbf{h}^N$ to get the output vectors $\mathbf{y}_1, \ldots, \mathbf{y}_L$.

# B  EXPERIMENTAL SETTINGS

## B.1  REASONING IN SYNTHETIC SETTINGS

In this experiment, we use $C_{out} = 10, L = 50$ for the model encoder. Each word is embedded with 50 dimension vectors in the decoder. The decoder has 2 layers, each multi-head attention module has

4 heads, and $W_i^Q, W_i^K, W_i^V \in \mathbb{R}^{50 \times 8}, W^O \in \mathbb{R}^{8 \times 50}$. The model is trained for $500$ epochs using Adam optimizer with initial learning rate at $0.001$ and a batch size is set as $32$.

To further examine the model's ability on compositionality task, we evaluate another version of the model which replaces the decoder with two linear transformations to directly predict $y_l \in\{$topleft, bottomright$\}$, and $y_c \in\{$Blue, Red, Color$\}$. With the hidden representation of the encoder **c** in equation 1, $y_l$ and $y_c$ are calculated as

$$y_l = \text{softmax}(\tilde{\mathbf{c}}W_l), \quad y_c = \text{softmax}(\tilde{\mathbf{c}}W_c) \tag{8}$$

where $W_l \in \mathbb{R}^{|\tilde{\mathbf{c}}| \times 2}, W_c \in \mathbb{R}^{|\tilde{\mathbf{c}}| \times 3}$, and $\tilde{\mathbf{c}}$ is the flattened vector of **c**.

### B.2 ESTIMATING THE DISTRIBUTION OF HUMAN QUESTIONS

In this experiment, the model encoder has the same hyper-parameters as in the first experiment. We increase the size of the decoder by setting number of layers to 3, number of heads to 6, and set $W_i^Q, W_i^K, W_i^V \in \mathbb{R}^{50 \times 32}, W^O \in \mathbb{R}^{32 \times 50}$. The model is also trained for $500$ epochs using Adam optimizer with the same initial learning rate at $0.001$. In order to better familiarize the model with grammar, we also pretrain the decoder for $500$ epochs on a larger set of question programs. This pretraining corpus is first uniformly sampled from the PCFG grammar defining questions, then we calculate the average energy value of each program on $10,000$ boards, and keep the top $2500$ unique questions.

For the evaluation set of human questions, we found that some simple questions become complicated in program form. For example, question "How many ship pieces are there in the second column?" will be translated to "(++ (map (lambda y (and (== (colL y) 2) (not (== (color y) Water)))) (set All_Tiles)))". Such complicated programs score very poorly according to the energy function, so they do not appear in the training set. As a result, the average log-likelihood is extremely skewed by these unseen questions. For a more robust evaluation, we remove the last $10\%$ human questions with low energy values according to Rothe et al. (2017).

### B.3 QUESTION GENERATION

The neural model for this experiment has the same hyper-parameters as in the last experiment, and is optimized by REINFORCE (Williams, 1992) algorithm with initial learning rate $0.0001$ and batch size $24$. To encourage the exploration of the model, we also apply an $\epsilon$-greedy strategy with $\epsilon$ set to $0.5$ at the beginning, and gradually decreased to $0.01$ as training continues. This model is trained for $500$ epochs, within each epoch the model passes $10,000$ different boards.

From some preliminary experiments, we find that the models have a strong preference of generating similar programs of relatively low complexity, with the original energy values as rewards. Thus, we tune two parameters of the energy model as mentioned in section 5.3, which are the two parameters corresponding to information seeking features (denoted as $f_{EIG}$ and $f_{EIG=0}$ in the original paper Rothe et al. (2017)). We increase this two parameters from $0.1$ until the reinforcement learning models are able to generate a diverse set of questions.

The sequence RL baseline which directly generates sequence with the decoder is trained with MIXER algorithm (Ranzato et al., 2016), which is a variant of REINFORCE algorithm widely used in sequence generation tasks. MIXER provides a smooth transition from supervised learning to reinforcement learning. This model is pretrained for $500$ epochs, and trained with RL for $300$ epochs.

## C  FULL GRAMMAR FOR BATTLESHIP

The grammar is provided in table 4 and 5.

Table 4: Part 1 of the grammatical rules. Rules marked with [b] have a reference to the Battleship game board (e.g., during evaluation the function *orient* looks up the orientation of a ship on the game board) while all other rules are domain-general (i.e., can be evaluated without access to a game board).

| | |
|---|---|
| **Answer types** | |
| A → B | *Boolean* |
| A → N | *Number* |
| A → C | *Color* |
| A → O | *Orientation* |
| A → L | *Location* |
| **Boolean** | |
| B → True | |
| B → False | |
| B → (not B) | |
| B → (and B B) | |
| B → (or B B) | |
| B → (== B B) | |
| B → (== N N) | |
| B → (== O O) | |
| B → (== C C) | |
| B → (=== setN) | *True if all elements in set of numbers are equal* |
| B → (any setB) | *True if any element in set of booleans is True* |
| B → (all setB) | *True if all elements in set of booleans are True* |
| B → (> N N) | |
| B → (< N N) | |
| B → (touch S S) [b] | *True if the two ships are touching (diagonal does not count)* |
| B → (isSubset setL setL) | *True if the first set of locations is subset of the second set of locations* |
| **Numbers** | |
| N → 0 | |
| ... | |
| N → 10 | |
| N → (+ N N) | |
| N → (+ B B) | |
| N → (++ setN) | |
| N → (++ setB) | *Number of True elements in set of booleans* |
| N → (− N N) | |
| N → (size S) [b] | *Size of the ship* |
| N → (row L) | *Row number of location L* |
| N → (col L) | *Column number of location L* |
| N → (setSize setL) | *Number of elements in set of locations* |
| **Colors** | |
| C → S | *Ship color* |
| C → Water | |
| C → (color L) [b] | *Color at location L* |
| S → Blue | |
| S → Red | |
| S → Purple | |
| S → x | *Lambda variable for ships* |
| **Orientation** | |
| O → H | *Horizontal* |
| O → V | *Vertical* |
| O → (orient S) [b] | *Orientation of the ship S* |
| **Locations** | |
| L → 1-1 | *Row 1, column 1* |
| ... | |
| L → 6-6 | |
| L → y | *Lambda variable for locations* |
| L → (topleft setL) | *The most left of the most top location in the set of locations* |
| L → (bottomright setL) | *The most right of the most bottom location in the set of locations* |

Table 5: Part 2 of the grammatical rules. See text for details.

**Mapping**
setB → (map fyB setL)          *Map a boolean expression onto location set*
setB → (map fxB setS)          *Map a boolean expression onto ship set*
setN → (map fxN setS)          *Map a numerical expression onto ship set*
setL → (map fxL setS)          *Map a location expression onto ship set*

**Lambda expressions**
fyB → (λ y B)                  *Boolean expression with location variable*
fxB → (λ x B)                  *Boolean expression with ship variable*
fxN → (λ x N)                  *Numeric expression with ship variable*
fxL → (λ x L)                  *Location expression with ship variable*

**Sets**
setS → (set AllShips)                    *All ships*
setL → (set AllTiles)                    *All locations*
setL → (coloredTiles C) [b]              *All locations with color C*
setL → (setDifference setL setL)         *Remove second set from first set*
setL → (union setL setL)                 *Combine both sets*
setL → (intersection setL setL)          *Elements that exist in both sets*
setL → (unique setL)                     *Unique elements in set*

Table 6: Results of the synthetic reasoning tasks.

(a) Accuracy of different models on the counting and missing tile tasks

| Model | Counting | Missing tile |
|---|---|---|
| Full model | 99.80% | 95.50% |
| 3x3 conv only | 99.30% | 51.50% |
| 1x1 conv only | 98.60% | 1.90% |

(b) Accuracy for selecting the right tile location and color for the missing tile

| Model | Location acc. | Color acc. |
|---|---|---|
| Full model | 97.80% | 97.60% |
| 3x3 conv only | 53.00% | 96.20% |
| 1x1 conv only | 3.90% | 45.50% |

# D   ADDITIONAL RESULTS

We also perform an ablation test on the neural network in the first experiment (section 5.1). Accuracy for the counting and missing tile tasks is summarized in Table 6a for the full model and lesioned variants. The full neural program generation model shows strong reasoning abilities, achieving an accuracy of 99.80% and 95.50% for the counting and missing tile tasks, respectively. The full model is compared to weakened variants with only one filter size in the encoder, either "3x3" and "1x1 conv only," and the performance of the weakened models drop dramatically on the missing tile task.

To better understand the role of different filter sizes, Table 6b breaks down the errors in the missing tile task on whether the question can pick the right ship (color acc. ) and whether it can select the right location (location acc.). The $3 \times 3$ convolution filters can accurately select the correct color, but often fail to choose the right tile. The model with $1 \times 1$ convolution filters has poor performance for both color and location. In the current architecture, predicting the correct location requires precise information that seems to be lost without filters of different sizes.

For the experiment on estimating the distribution of human questions (Experiment 5.2), Table 7 provides a full table of log-likelihood of different models on evaluation set of different uncertainty level.

Here we provide more examples of questions generated by our models in the generation experiment (Experiment 5.3). Figure 5, 6 and 7 provides more examples for the same settings as shown in figure 4 in the main text. Figure 8 shows generated examples of the text-based model.

Table 7: Log-likelihood (LL) on different splits of the sampled evaluations based on the uncertainty of the board.

| Model | LL on low entropy | LL on mid entropy | LL on high entropy |
|---|---|---|---|
| Full model | **-2.990** | **-3.190** | **-3.414** |
| no pretrain | -3.015 | -3.210 | -3.428 |
| LSTM decoder | -3.044 | -3.209 | -3.416 |
| MLP encoder | -3.293 | -3.383 | -3.477 |
| decoder only | -3.312 | -3.397 | -3.494 |

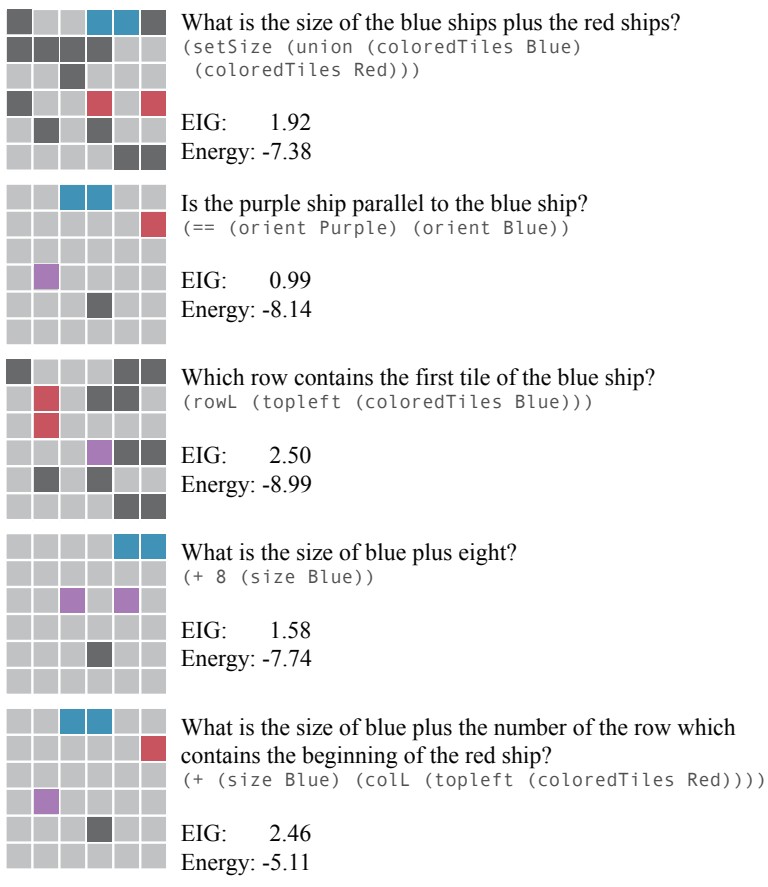

Figure 5: Novel questions generated by the grammar enhanced model.

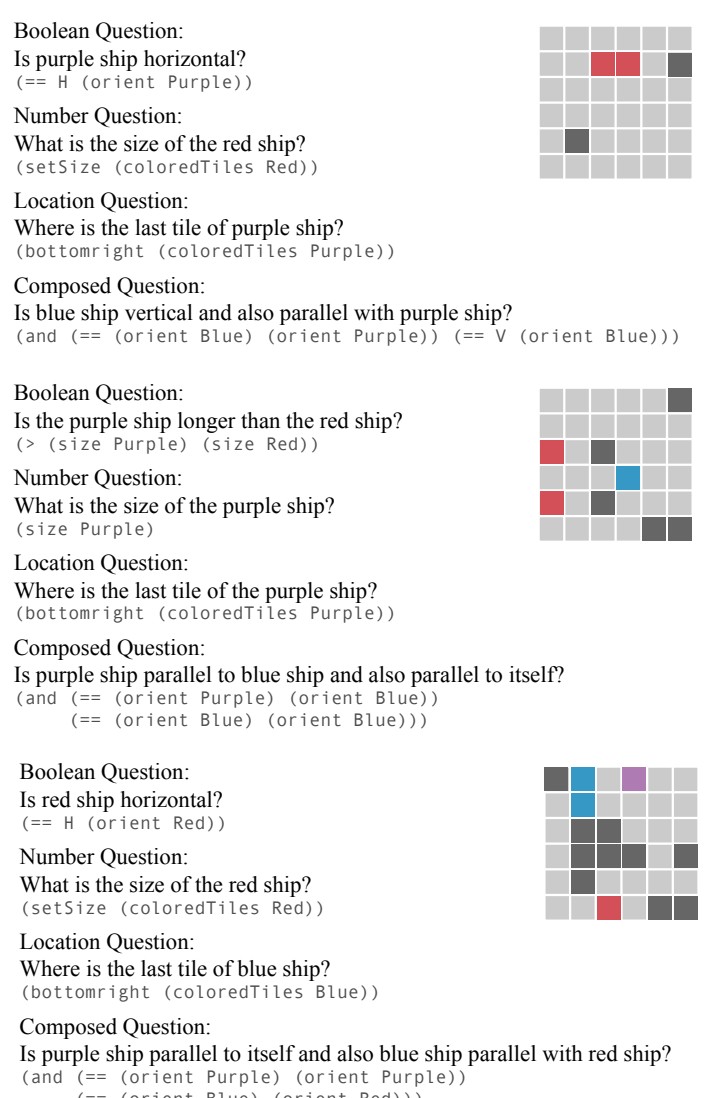

Boolean Question:
Is purple ship horizontal?
(== H (orient Purple))

Number Question:
What is the size of the red ship?
(setSize (coloredTiles Red))

Location Question:
Where is the last tile of purple ship?
(bottomright (coloredTiles Purple))

Composed Question:
Is blue ship vertical and also parallel with purple ship?
(and (== (orient Blue) (orient Purple)) (== V (orient Blue)))

Boolean Question:
Is the purple ship longer than the red ship?
(> (size Purple) (size Red))

Number Question:
What is the size of the purple ship?
(size Purple)

Location Question:
Where is the last tile of the purple ship?
(bottomright (coloredTiles Purple))

Composed Question:
Is purple ship parallel to blue ship and also parallel to itself?
(and (== (orient Purple) (orient Blue))
     (== (orient Blue) (orient Blue)))

Boolean Question:
Is red ship horizontal?
(== H (orient Red))

Number Question:
What is the size of the red ship?
(setSize (coloredTiles Red))

Location Question:
Where is the last tile of blue ship?
(bottomright (coloredTiles Blue))

Composed Question:
Is purple ship parallel to itself and also blue ship parallel with red ship?
(and (== (orient Purple) (orient Purple))
     (== (orient Blue) (orient Red)))

Figure 6: Generated questions of different types by controlling the start condition.

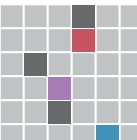

Our model:
Where is the bottom right of the blue ship?
`(bottomright (coloredTiles Blue))`
What is the size of the blue ship?
`(setSize (coloredTiles Blue))`
What is the orientation of the blue ship?
`(orient Purple)`
Where is the bottom right of the purple ship?
`(bottomright (coloredTiles Purple))`
What is the size of the blue ship?
`(size Blue)`
What is the size of the blue ship?
`(setSize (coloredTiles Blue))`

Human:
Are the majority of the ships horizontal or vertical?
`(> (++ (map (lambda x (== (orient x) H))`
`   (set AllColors))) 1)`
Which direction is purple?
`(== (orient Purple) H)`
Is the red ship placed vertically?
`(== (orient Red) H)`
3 tiles is blue ship?
`(== (size Blue) 3)`
Does tile B6 hold a part of the blue ship?
`(== (color 6-2) Blue)`
Where is the end of the blue ship
`(bottomright (coloredTiles Blue))`

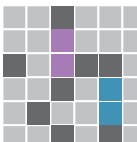

Our model:
What is the size of the blue ship?
`(size Blue)`
Is the red ship in parallel with the blue ship?
`(== (orient Red) (orient Blue))`
What is the size of the red ship plus 5?
`( + 5 ( setSize ( coloredTiles Red ) ) )`
What is the size of the red ship?
`( setSize ( coloredTiles Red ) )`
What is the orientation of the red ship?
`( orient Red )`
Where is the bottom right of the blue ship?
`( bottomright ( coloredTiles Blue ) )`

Human:
Is the red ship touching another ship?
`(or (touch Red Blue) (touch Red Purple))`
Is there an item at 6-1 tile?
`(not (== (color 6-1) Water))`
How many tiles is the red ship?
`(size Red)`
Is the red ship touching another ship?
`(or (touch Red Blue) (touch Red Purple))`
Is orange ship 2 tiles long?
`(== (size Red) 2)`
Is any part of the red ship in left half of the grid?
`(> (++ (map (lambda y (and (< (colL y) 4)`
`   (== (color y) Red))) (set AllTiles))) 0)`

Figure 7: Comparisons of questions generated by our model with human questions.

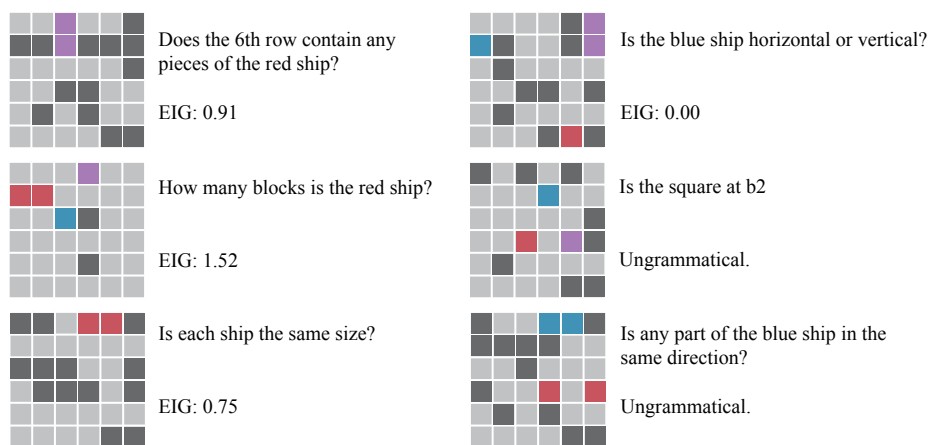

Figure 8: Example questions generated by the text-based model.

