# OpenReview forum: "Modeling question asking using neural program generation"
_ICLR.cc/2020/Conference — Reject_

### Official Review · AnonReviewer1 · 2019-10-15
**Official Blind Review #1**

**Rating:** 1

**Review:**

The authors explore different ways to generate questions about the current state of a “Battleship” game. To do this, they introduce a neural network architecture with a convolutional encoder and a Transformer-based decoder and consider both supervised and reinforcement-learned training approaches. They evaluate the introduced methods in three different ways that demonstrate basic classification and generation abilities of the model.

While the paper is generally well written and clear, I noticed quite a few grammar mistakes and typos. I expect these can be fixed in the final version.


In my opinion this work has several limitations and I don’t think it is a strong enough contribution to be interesting to the ICLR audience. Thus I am leaning towards rejection.


While the problem of question generation is quite interesting, this work is limited in several ways. Firstly the authors limit the domain of question answering severely by just considering questions about “Battleship” that can be expressed in a lisp-like language. They do not attempt to generate questions that are oriented towards a more complex goal (like winning the game), but instead use three tasks that are quite limited:

- First they show the model works well on very simple toy tasks; in particular the tasks are 1. Identifying the colour with the least visible tiles, 2. Identifying the ship with a missing tile, and 3. Identifying the colour of the ship with both a missing tile and the least number of visible tiles. These are extremely simple classification problems and no real question has to be generated. While this is a good sanity check for the model, it is not an interesting result.
- In a second task they try to show that the model can capture the distribution of human authored questions. However, they do not use human-authored questions but instead use importance sampling using a heuristic method from Rothe et al. My skepticism of this experiment comes from the fact that  the “decoderless model” that does not take into account the board state performs similar to models that do (and even outperforms the model with LSTM decoder). I am not sure how meaningful this task is.
- Lastly they show that using REINFORCE with a tweaked reward (a version of the energy function from Rothe et al.) produces new kinds of questions that have not seen before in the training data. This is not surprising and also not very meaningful without either an application or a thorough qualitative analysis of the generated questions.

So while extending the work of Rothe et al. with neural architectures seems like an interesting endeavour, the tasks the authors use in the paper are severely limited and do not demonstrate interesting behaviour.


I believe this is not in scope for changes to this paper, but in general it could be interesting to evaluate the question generation in an end-to-end setup, similar to the reformulation framework used by Nogueira and Cho in “Task-Oriented Query Reformulation with Reinforcement Learning” and Buck et al. in “Ask the Right Questions: Active Question Reformulation with Reinforcement Learning”.

Nit: The colours chosen for the diagrams could be improved. In particular purple is quite hard to distinguish from the dark gray.


**Experience Assessment:**

I have published one or two papers in this area.

**Review Assessment: Checking Correctness Of Derivations And Theory:**

I carefully checked the derivations and theory.

**Review Assessment: Checking Correctness Of Experiments:**

I carefully checked the experiments.

**Review Assessment: Thoroughness In Paper Reading:**

I read the paper thoroughly.

---

> ### Author Response · Authors · 2019-11-15
> **Response to review #1**
>
> We sincerely thank you for your feedback, and we are glad you found question generation to be an interesting challenge. The crux of your critique concerns the choice of tasks. We see the Battleship domain as an ideal testbed for developing computational approaches to question asking. The domain offers a clear goal (finding the ships) and the opportunity to ask natural language questions in a relatively unconstrained setting. At the same time, questions can be compared quantitatively using ideal Bayesian measures of value, and in some of the settings (that we study) the optimal question can be defined. While everyone is entitled to their opinion about what interests them, we respectfully disagree with your comment that our tasks “do not demonstrate interesting behaviour” and below we review the various behavioral demonstrations that we found to be highly interesting.
>
> Most question asking tasks concern “chit chat” questions that are not goal directed and do not permit quantitative measures of question value. Our tasks allow us to define the optimal question in certain settings. Our first experiment demonstrates that our neural program generation model can learn to ask optimal questions, and we also study our model’s ability to generalize to novel compositions of functions and arguments.
>
> The second experiment addresses question asking in an unconstrained setting of density estimation, based on a set of human-like questions. Although we incorporate human generated questions, the amount of behavioral data needed for naive supervised training is prohibitive. To overcome this limitation,  we designed a data augmentation procedure that leverages a previous model from Rothe et al. (2017) to produce additional human-like questions. We now better explain this choice and highlight it in the introduction (see revised Introduction). Using this dataset, we find good predictive performance compared to a range of baselines.
>
> The third experiment uses reinforcement learning (RL) to learn to ask good questions in a more autonomous manner, further minimizing the need for many human-generated questions. We explored how RL and context-free grammars can be combined in a novel way to ensure the grammatical validity, diversity, and informativeness of the generated programs without human question demonstrations. We find the system generates novel questions such as “Where is the bottom right of all the purple and blue tiles together?” and other examples mentioned in the paper.
>
> Regarding the second experiment, you mentioned concerns that a model without encoder performs similarly to other models. As we explain in the paper and further clarify in our revision (see revised Section 5.2.2), some questions are strong across a wide range of contexts. For example, questions like “how long is the blue ship” might be a good question to ask in most cases, and the decoder can capture these common questions without requiring an encoder. This is why we split the dataset by entropy to study the role of context in more detail (Table 2b), finding models without an encoder perform much worse when entropy is low. Also the first experiment would not be possible without an encoder.
>
> We agree that allowing the agent to play a complete game is an interesting future direction. With reinforcement learning, it is possible to extend our framework and allow the agent to iteratively ask questions and try to win the game; it’s possible that we can even use the end result as the only source of reward. But we wish to emphasize that the single-turn, goal-directed question asking task itself is very important -- especially in a partial-observable setting -- and is currently under-explored in the literature. Question asking in partially observed settings is important to human interaction in daily life, and also sets the stage for practical applications. For a personal assistant, an agent needs to ask questions to quickly understand the intentions of users. In educational settings, teachers ask questions to infer the mental states of students. These are high-value scenarios for AI research and applications, and are important future directions of our work.
>
> For a minor point, we adjusted the colors in the figures. Thank you for your suggestion.

---

### Official Review · AnonReviewer2 · 2019-10-16
**Official Blind Review #2**

**Rating:** 6

**Review:**

The paper uses a deep neural network architecture (CNN + Transformer) to model logical translations of questions in the form of programs. The experimental setup uses the "battleship" game scenario, which is an interesting domain for questions because of the inherent partial observability present in the game. The paper is clearly written and well-presented.

The length of the submission is 9 pages in content. The call for papers states that "Reviewers will be instructed to apply a higher standard to papers in excess of 8 pages." Given the major reservation I have regarding the current version (see below), and the need to apply a higher standard, it is hard for me to recommend acceptance in its current form.

The major reservation I have is that the paper currently sets up the reader to expect the data to contain real, human-generated questions. The abstract mentions "human question asking", talks about predicting "which questions humans are likely to ask in unconstrained settings"; on p.2 the paper states that "the model can be trained from human demonstrations of good questions"; and so on. Hence it came as a surprise to learn on p.6 that collecting human questions and translating them into programs is too laborious, so an automatic system was used to generate the questions instead. This isn't necessarily a killer for the paper, but it does require more justification, and more sign-posting early in the paper.

On p.3 there is some attempt at justifying the decision to use a simulator, which is that an existing paper has shown that it captures the full range of questions that people ask. I feel that this justification needs expanding.

Another issue with the paper is that it heavily builds on the existing work by Rothe et al., in fact so much so that it's difficult to fully appreciate the current paper without reading that existing work (which I haven't done, just for full disclosure). In fact, a summary of the current paper would be that it takes the existing work of Rothe et al., which is a rule-based question-generation system, and it "neuralizes" it by replacing the rules with a neural architecture.

That said, I think there is still a lot to commend in this work: the setting is an interesting one for question generation, the motivation for using programs is well-made, and the work appears technically sound.

Some more minor comments
--

I wonder about the log-likelihood numbers on the synthetic questions. Doesn't this just show that a neural network can effectively reverse-engineer a synthetic grammar? I'm not sure how interesting that is. I also wonder about the decision to filter questions that score poorly according to the generative model used for training the network - doesn't this just add additional bias in favour of the model being evaluated?

A similar comment applies to the decision to exclude ungrammatical questions from the uniqueness metric. I assume "ungrammatical" here means ungrammatical according to the pre-defined grammar. But that just makes it easier for the grammar-based system to perform well on this particular metric, no?

I don't think EIG is defined anywhere in the paper.


**Experience Assessment:**

I have published in this field for several years.

**Review Assessment: Checking Correctness Of Derivations And Theory:**

I assessed the sensibility of the derivations and theory.

**Review Assessment: Checking Correctness Of Experiments:**

I assessed the sensibility of the experiments.

**Review Assessment: Thoroughness In Paper Reading:**

I read the paper thoroughly.

---

> ### Author Response · Authors · 2019-11-15
> **Response to review #2**
>
> We are pleased that you found the domain and challenge interesting, and you found the work well-presented and technically sound. Thank you for your thoughtful suggestions for improvements.
>
> As you rightly point out, our work builds upon Rothe et al. (2017). We use this domain as an ideal testbed for developing models of question asking (see our response to review #1), and their paper was first to study goal-directed, question generation from a partially-observable state. We also build upon Rothe et al. ’s (2017) program-based representation language for questions. However, we see our “neural program generation” framework as an important advance. The method of Rothe et al. (2017) is not a generation-oriented method; it relies on hand-crafted features and it is extremely computationally expensive to generate a question using their model. Our neural program generation approach can quickly generate new questions, and can even learn to ask good questions through reinforcement learning without supervision, broadening the potential applications.
>
> You mentioned that "the major reservation I have is that the paper currently sets up the reader to expect the data to contain real, human-generated questions”.  We would like to thank you for your advice and revised our paper to better explain this choice, as well as mention it in the introduction (see revised Introduction and Section 5.2.1). In sum, although we incorporate human generated questions, the amount of behavioral data needed for naive supervised training is prohibitive. To overcome this limitation, we designed a data augmentation procedure that leverages a previous model from Rothe et al. (2017) to produce additional human-like questions. Their model uses hand-crafted features to estimate the probability of questions being asked by humans, but it requires a very slow search procedure. Offline, we train their model on real human dataset and sample a large number of human-like questions as augmented data. Then we assessed our method in an unconstrained setting with density estimation, based on these human-like questions. Given the difficulty of collecting a massive amount of human questions and manually translate them into programs with expert knowledge, we believe our current approach is the best alternative.
>
> As for filtering out some human questions, we see this primarily as a limitation of the grammar itself and its ability to model very long questions. Any model that uses the grammar and the data-augmentation procedure would be influenced similarly, thus it is still a fair comparison with the alternatives in our paper. For the third experiment, grammatically invalid questions should not be considered novel questions because they are not semantically coherent.
>
> We also added the definition and formula for EIG. Thanks for pointing out this omission.
>
> Finally, you mentioned that since there were 9 pages of content, there was “the need to apply a higher standard, it is hard for me to recommend acceptance in its current form.” We apologize for the inconvenience regarding the longer initial version. We spent considerable efforts to shorten the paper to 8 pages of content. We kindly ask that you evaluate our paper in light of these revisions.

---

### Official Review · AnonReviewer3 · 2019-10-23
**Official Blind Review #3**

**Rating:** 6

**Review:**

[Summary]
This paper studies a very interesting problem - if machines can learn to ask the right questions to address or gain crucial information about tasks. I believe this is a very important yet under-explored problem. Specifically, this paper considers a setting where a neural network model is trained to ask questions by predicting a formal program. The network consists of a CNN encoder which encodes a partially observable state and a Transformer decoder that generates a program as a sequence of tokens. The experiments on a Battleship task show some promising results.

Significance: are the results significant? 3/5
Novelty: are the problems or approaches novel? 3/5
Evaluation: are claims well-supported by theoretical analysis or experimental results? 4/5
Clarity: is the paper well-organized and clearly written? 4.5/5

[Strengths]

*motivation*
The motivation for investigating machines' ability to generating questions to gain important information is convincing.

*novelty*
The idea of utilizing learning models for generating questions and modeling human-generated questions intuitive and convincing. This paper presents an effective way to implement this idea.

*technical contribution*
Leveraging a context-free grammar (CFG) and reinforcement learning (RL) for question generation seems effective especially when it comes to generating unique and novel questions.

*clarity*
The overall writing is clear and the authors utilize figures well to illustrate the ideas. Figure 1 illustrates the Battleship task and Figure 2 presents a clear overview of the proposed framework.

*ablation study*
Ablation studies are comprehensive. The proposed framework consists of multiple components. The provided ablation studies (Table 2 and Table 4) help analyze the effectiveness of each of them.

*experimental results*
- The presentations of the results are clear. The conclusions are fairly convincing:
- Experiment 1: learning to generate programmatic questions encourages learning rules for composition
- Experiment 2: it is possible to capture the human-generated questions distribution as a conditioned language model.
- Experiment 3: CFG+RL can produce more diverse and novel questions compared to supervised learning or RL.

[Weaknesses]

*novelty & contribution*
Overall, I do not find enough novelty from any aspects while the overall effort of this paper is appreciated.
- The problem is not entirely novel as [1-3] have explored asking questions with neural networks learning to generate programmatic questions.
- The proposed framework leverages a CFG and learns with RL, while the former (program synthesis with a CFG) has been studied in [4-5] and the latter (program synthesis using RL) has been discussed in [4, 6-7].
- Many setups considered in this paper have been explored in several neural program synthesis papers [8-12], which are neglected from this paper.
- The conclusions presented in the experiment section are more or less expected (i.e. other works have presented similar results on slightly different problems).

*oversell the motivation*
I believe the authors oversell the key motivation (i.e. to enable learning agents to ask questions) a little bit. To be more specific, it would be more interesting if an agent is allowed to ask a question, take actions based on the answer, and then ask the next question. However, this paper investigates the case where a significant amount of human-generated questions are given, which could limit the learning in my opinion. Also, it is not clear to me how the setup evaluated in this paper can be extended to a setting that allows an agent to iteratively ask questions.

*experiment setup*
Do the models learning using RL get the reward based on just the ground truth questions (a sequence of tokens) or the execution results?

*format*
The title & reference format looks wrong

*reference*
[1] Question Asking as Program Generation in NIPS 2017
[2] Neural-Symbolic VQA: Disentangling Reasoning from Vision and Language Understanding in NeurIPS 2018
[3] The Neuro-Symbolic Concept Learner: Interpreting Scenes, Words, and Sentences From Natural Supervision in ICLR 2019
[4] Leveraging grammar and reinforcement learning for neural program synthesis in ICLR 2018
[5] Learning a Meta-Solver for Syntax-Guided Program Synthesis in ICLR 2019
[6] Neural Scene De-rendering in CVPR 2017
[7] Seq2SQL: Generating Structured Queries from Natural Language using Reinforcement Learning
[8] RobustFill: Neural Program Learning under Noisy I/O in ICML 2017
[9] Execution-Guided Neural Program Synthesis in ICLR 2019
[10] Neural Program Synthesis from Diverse Demonstration Videos  in ICML 2018
[11] Learning to Describe Scenes with Programs in ICLR 2019
[12] Learning to Infer and Execute 3D Shape Programs in ICLR 2019

**Experience Assessment:**

I have published one or two papers in this area.

**Review Assessment: Checking Correctness Of Derivations And Theory:**

N/A

**Review Assessment: Checking Correctness Of Experiments:**

I assessed the sensibility of the experiments.

**Review Assessment: Thoroughness In Paper Reading:**

I read the paper at least twice and used my best judgement in assessing the paper.

---

> ### Author Response · Authors · 2019-11-15
> **Response to review #3**
>
> Thank you for your review, and we are glad that you find our domain to be interesting and a “very important yet under-explored problem.” In light of your comments, we would like to comment more extensively on the novelty of our work and how we revised our introduction in response.
>
> You mention that “the problem is not entirely novel as [1-3] have explored asking questions with neural networks learning to generate programmatic questions.” We feel it’s important to distinguish our work on *question asking* (also covered in [1]) from the other two papers you cite which address only *question answering* [2-3]. Indeed, we thank you for the additional references (many of which we include in our revision; see revised Introduction and Related work), but it’s important to mention that these suggestions for related work address either question answering or problem solving, not question asking [2-12]. Question answering and question asking are related but inherently different. Question answering is much more straightforward because there is a ground-truth answer for each question. In contrast, question asking requires a search through infinitely many questions where many questions may be appropriate and informative. Question answering systems, especially models in VQA settings, usually optimize accuracy given a set of provided answers. For question asking, it is much harder to define a good objective function.
>
> The prior work on question asking, although much less frequently studied than question answering, usually addresses a very different kind of task. Neural question asking has focused on the open-ended, reading comprehension style questions, and the questions are generated using a fully observable state [see references 13-15 below]. In the Battleship setting, a successful agent must ask information-seeking questions from a partially observable state, utilizing compositionality to synthesize questions that are fitting in the scenario.
>
> Our framework addresses these challenges by modeling human question asking as a combination of symbolic programs and neural networks. Compositional programs explain the flexibility and creativity question formation, while neural networks can quickly synthesize questions that are sensitive to the partially-revealed world state. Although prior work has explored either grammars or RL in program synthesis, few papers have found ways to integrate both successfully. A common strategy is to simply filter syntactically invalid programs during generation, and/or discourage them with a negative reward in RL. The most closely related paper to our combines a grammar and RL [4,5] by encoding grammar with an auxiliary module like LSTM in [4] and GNN in [5].
>
> Importantly, our aim is not to propose a new method for neural program synthesis in any general sense. Our paper introduces a framework for question asking in goal-oriented, information-seeking, partially observable scenarios. The paper of Rothe et al. [1] is the first paper to tackle this kind of domain, and our paper presents a substantial advance over that method. Their method is not a generation-oriented method, it relies on many hand-crafted features and uses a simple linear model to estimate the probability of a question being asked. It is very time-consuming to generate a question using their model, which a large number of questions need to be sampled and evaluated to ensure the quality of generated question. Our neural method avoids this issue, and is more practical in applications. To solve the problem that deep neural networks are data hungry, we proposed two solutions. One is the data augmentation schema we described in Section 5.2, the other is to use RL. And by this novel overall framework, we find a possible solution to the question asking task under partially observable states.
>
> We agree that allowing the agent to play multiple rounds or a complete game is an interesting future direction which we added in the revised paper (see Conclusion). With reinforcement learning, it is possible to extend our framework and allow the agent to iteratively ask questions and try to win the game; it’s possible that we can even use the end result as the source of reward. But we wish to emphasize that the single-turn, goal-directed question asking task itself is very important -- especially in a partial-observable setting -- and is currently under-explored (as you mentioned in your review) in the literature.
>
> For other minors points you mentioned, the models trained with RL get rewards based on execution results. And thank you for pointing out the format error, we have fixed the error in the new revision.
>
> In addition to your list [1-12], we add three additional references for our reply:
> [13] Ask the Right Questions: Active Question Reformulation with Reinforcement Learning in ICLR 2018
> [14] Teaching machines to ask questions in IJCAI 2018
> [15] Generating Natural Questions About an Image in ACL 2016

---

> > ### Comment · AnonReviewer3 · 2019-11-15
> > **Re: Response to review #3**
> >
> > I appreciate the authors for the response. I agree that program synthesis and questions asking are very different. I mainly brought up those program synthesis works simply because all this paper and those works aim to generate programs and I believe some of the techniques proposed in those works could potentially be applied here. I do not have any further questions at this moment.

---

### Author Response · Authors · 2019-11-15
**General response to reviewers**

Dear reviewers,

We thank each of you for your thoughtful feedback on our work. We have carefully considered your reviews and incorporated your suggestions into a revised copy of our paper, which has now been uploaded. We explain the specifics of our revision in comments directly appended to each review. We want to highlight that we reduced the length of the paper to fit within 8 pages. Please see our specific responses below, and thank you for considering our work for ICLR in your ongoing discussions.

Sincerely,
The authors

---

### Decision · Program_Chairs · 2019-12-19

**Decision:**

Reject

**Comment:**

The authors explore different ways to generate questions about the current state of a “Battleship” game. Overall the reviewers feel that the problem setting is interesting, and the program generation part is also interesting. However, the proposed approach is evaluated in tangential tasks rather than learning to generate question to achieve the goal. Improving this part is essential to improve the quality of the work.